# Measuring Spinal Mobility Using an Inertial Measurement Unit System: A Validation Study in Axial Spondyloarthritis

**DOI:** 10.3390/diagnostics10060426

**Published:** 2020-06-24

**Authors:** I. Concepción Aranda-Valera, Antonio Cuesta-Vargas, Juan L. Garrido-Castro, Philip V. Gardiner, Clementina López-Medina, Pedro M. Machado, Joan Condell, James Connolly, Jonathan M. Williams, Karla Muñoz-Esquivel, Tom O’Dwyer, M. Carmen Castro-Villegas, Cristina González-Navas, Eduardo Collantes-Estévez

**Affiliations:** 1Faculty of Medicine, University of Córdoba, 14005 Córdoba, Spain; conchita.87.8@gmail.com (I.C.A.-V.); mcasvi@yahoo.es (M.C.C.-V.); crsgonzaleznavas@yahoo.es (C.G.-N.); ecollantes@yahoo.es (E.C.-E.); 2Rheumatology Department, University Hospital Reina Sofía, 14005 Córdoba, Spain; clementinalopezmedina@gmail.com; 3Maimonides Biomedical Research Institute of Cordoba, 14005 Córdoba, Spain; 4Physiotherapy Department IBIMA, University of Malaga, 29010 Malaga, Spain; acuesta@uma.es; 5Computing and Numerical Analysis Department, University of Cordoba, 14014 Córdoba, Spain; 6Western Health and Social Care Trust, Londonderry BT47 6SB, UK; philip.gardiner@westerntrust.hscni.net; 7Department of Rheumatology, University College London Hospital NHS Foundation Trust, London NW1 2PG, UK; p.machado@ucl.ac.uk; 8Intelligent Systems Research Centre, University of Ulster, Derry BT48 7JL, UK; j.condell@ulster.ac.uk (J.C.); kc.munoz-esquivel@ulster.ac.uk (K.M.-E.); 9Letterkenny Institute of Technology, F92 FC93 Letterkenny, Ireland; james.connolly@lyit.ie; 10Department of Rehabilitation and Sports Sciences, Faculty of Health and Social Sciences, Bournemouth University, Bournemouth BH12 5BB, UK; jwilliams@bournemouth.ac.uk; 11Independent Researcher, D08 W9RT Dublin, Ireland; odwyertk@tcd.ie

**Keywords:** axial spondyloarthritis, spinal mobility, inertial measurement unit

## Abstract

Portable inertial measurement units (IMUs) are beginning to be used in human motion analysis. These devices can be useful for the evaluation of spinal mobility in individuals with axial spondyloarthritis (axSpA). The objectives of this study were to assess (a) concurrent criterion validity in individuals with axSpA by comparing spinal mobility measured by an IMU sensor-based system vs. optical motion capture as the reference standard; (b) discriminant validity comparing mobility with healthy volunteers; (c) construct validity by comparing mobility results with relevant outcome measures. A total of 70 participants with axSpA and 20 healthy controls were included. Individuals with axSpA completed function and activity questionnaires, and their mobility was measured using conventional metrology for axSpA, an optical motion capture system, and an IMU sensor-based system. The UCOASMI, a metrology index based on measures obtained by motion capture, and the IUCOASMI, the same index using IMU measures, were also calculated. Descriptive and inferential analyses were conducted to show the relationships between outcome measures. There was excellent agreement (ICC > 0.90) between both systems and a significant correlation between the IUCOASMI and conventional metrology (*r* = 0.91), activity (*r* = 0.40), function (*r* = 0.62), quality of life (*r* = 0.55) and structural change (*r* = 0.76). This study demonstrates the validity of an IMU system to evaluate spinal mobility in axSpA. These systems are more feasible than optical motion capture systems, and they could be useful in clinical practice.

## 1. Introduction

Axial spondyloarthritis (axSpA) is a chronic rheumatic disease that is characterized by inflammatory and structural changes in the axial skeleton [1]. The clinical presentation is characterized by inflammatory back pain, morning stiffness, and fatigue, which lead to loss of spinal mobility and physical function impairment [2,3,4]. Spinal mobility impairment has been described as one of the most important outcomes in axSpA and has been included in the Assessment of SpondyloArthritis international Society (ASAS) core set of domains for symptom-modifying antirheumatic drugs, disease controlling antirheumatic therapy, physical therapy and clinical record keeping [5]. The association between spinal mobility impairment and radiographic changes in the spine has been widely demonstrated at the group level, while this association is not as evident at the individual level [6,7]. Recently, reference intervals of spinal mobility in healthy individuals have been described with the aim of determining unusual or extreme measurements [8].

Most of the accepted and validated mobility measures for axSpA are based on objective outcomes focused on the lumbar and cervical spine [9]. In fact, the most common index used for mobility evaluation is the Bath Ankylosing Spondylitis Metrology Index (BASMI), which includes lateral spinal flexion (LSF), the modified Schober’s test (mSchober), cervical rotation, the tragus-to-wall distance (TTW) and the intermalleolar distance (IMD) [10]. However, this approach has demonstrated high intra- and inter-observer variability and low responsiveness [11,12].

New technological tools based on the use of motion capture have been developed and validated with the aim of improving the reliability, accuracy, and responsiveness of spinal mobility assessments. One of these new instruments is the UCOTrack [13], an optical motion capture system validated in patients with axSpA. This method uses reflective markers located in certain anatomical locations of the patient, complemented by four infrared cameras. In this way, the system calculates the 3D positions of markers and estimates distances and angles by defining segments between markers. On the basis of this technique, the University of Cordoba Ankylosing Spondylitis Metrology Index (UCOASMI) was developed [14], which demonstrated higher levels of accuracy, reliability, and responsiveness than the BASMI in a multi-centre study [15]. However, because the UCOTrack requires a specific environment and time for data processing, feasibility issues make its use difficult in daily clinical practice.

In recent years, new tools able to measure movement have emerged, such as portable inertial measurement units (IMUs). This technology is beginning to be used in classic applications of motion capture, such as in gait analysis [16,17,18] or in the assessment of trunk [19,20,21,22] and cervical mobility [23]. The versatility, low cost, and size mean that assessments can extend beyond the clinic, including self-assessments of mobility, which can be shared remotely with the clinician [24]. Some of these types of sensors have been tested in axSpA to analyze the level of physical activity of individuals [25].

The utility of IMU sensor-based technology for motion capture is progressing rapidly. Some studies compare agreement between motion capture systems, with good results [26,27,28,29]. IMU sensor-based systems have also been used not only to obtain kinematic data, but also kinetic estimation of the forces acting across joints and vertebrae [30].

Among several clinically approved motion analysis systems based on inertial sensors, the ViMove system (Dorsavi©, Melbourne, Australia) has been used to test mobility in the lumbar spine region both in normal subjects and in those with chronic back pain [31,32,33]. We have recently published the results of an initial study testing intra- and inter-observer reliability in 40 axSpA participants using this sensor-based system [34]. In this study, we demonstrated that the IMU sensor-based system was reliable, but more studies are needed to demonstrate its validity.

Following the recommendations of the Outcome Measures in Rheumatology (OMERACT) and COnsensus-based Standards for the selection of health Measurement Instruments (COSMIN) initiatives [35], which recommend that measuring instruments must prove to be truthful (valid), discriminative, and feasible, we conducted this study aiming to (a) analyze concurrent criterion validity in individuals with axSpA by comparing the spinal mobility results from an IMU sensor-based system vs. motion capture; (b) analyze discriminant validity by hypothesis testing comparing mobility results with healthy volunteers; (c) to analyze construct validity by comparing the IMU sensor-based system results with relevant outcome measures and conventional metrology in individuals with axSpA.

## 2. Materials and Methods

### 2.1. Study Population

This is a clinical measurement study with a focus on concurrent, discriminant, and construct validity. We conducted the study with different groups formed by participants with axSpA, according to the ASAS classification criteria [36], and healthy people as a control group. The inclusion criteria were as follows: older than 18 years, diagnosis of axSpA, and clinically stable according to the opinion of a rheumatologist without treatment modifications in the previous three months. People who had disc disease, who had undergone previous surgery, or who were pregnant were excluded from the groups.

Three groups of participants were defined: (a) a first group of individuals with axSpA specially recruited for this study—axSpA-1; (b) a second group of healthy participants, age and sex-matched with the participants of the first group—control; (c) axSpA participants from the CASTRO (Córdoba Axial Spondyloarthritis Task force, Registry, and Outcomes Spondyloarthritis Registry) cohort—axSpA-2 (Figure 1).

AxSpA-1 participants were consecutively selected from the outpatient Department of Rheumatology and healthy participants were recruited from the hospital and the university, as well as via the researchers’ personal contacts. All of them signed a consent form, and the protocol was approved by the Ethics Committee of the Reina Sofia University Hospital from Cordoba (Spain) (Ref. 0887-N-17).

### 2.2. Data Collection and Assessment

Sociodemographic (age and sex) and anthropometric (weight, height, and body mass index—BMI) data were collected by the same observer (C.G.-N.) at the study visit. Measurements for conventional metrology were obtained according to the ASAS handbook [9]: LSF, mSchober, cervical rotation, TTW and IMD, as well as the BASMI.

Participants in the axSpA group completed physical function (Bath Ankylosing Spondylitis Function Index—BASFI) [37], disease activity (Bath Ankylosing Spondylitis Disease Activity Index—BASDAI) [12], global assessment (Bath Ankylosis Spondylitis Patient Global score—BASG) [38] and quality of life (Assessment of SpondyloArthritis International Society Health Index—ASAS HI) [39] questionnaires. For the axSpA-2 group, lateral radiographs of the cervical and lumbar spine were evaluated by a trained rheumatologist (I.C.A.-V.) and were scored for structural changes according to the modified Stoke Ankylosing Spondylitis Spine Score (mSASSS) [40], and laboratory tests for C-reactive protein to calculate the ASAS Disease Activity Score (ASDAS) [41] were completed. Anthropometric and mobility measures were performed at the same time of day and by the same physician for all participants to avoid variability in the measurements, especially those related to morning stiffness.

### 2.3. Instrumentation

Spinal mobility was measured with an IMU sensor-based system in all participants. In the axSpA-1 group, spinal mobility was also assessed using a motion capture system for the concurrent validity analysis.

The UCOTrack (iSAB©, Córdoba, Spain), is an optical motion capture system that records kinematic measures using reflective markers placed on the patient in standard anatomical locations. This system has been previously used in axSpA [13] to generate a spinal mobility score (using cervical and lumbar measures), referred to as the University of Cordoba Ankylosing Spondylitis Metrology Index (UCOASMI), which has been validated in axSpA patients [14]. The UCOASMI composite score is calculated using 15 markers that record five parameters: Cervical flexion, cervical rotation, lumbar flexion, shoulder and hip lateral flexion, and lumbar rotation. This score varies from 0 to 10 (better to worse mobility), is obtained from a selection of individual measurements based on their metric properties, and is calculated as a weighted average. This method was used as the reference to determine the concurrent validity of the IMU sensor-based system.

The ViMove system (Dorsavi©), which uses portable inertial sensors based on IMUs, allows the measurement of cervical and lumbar mobility (flexion/extension, lateral flexion, and rotation) and was used as an IMU sensor-based system. Cervical movement was evaluated with two sensors: The upper sensor was attached using a head strap, which enabled the positioning of the sensor on the occiput, and the lower sensor was positioned at the T3 level by manual palpation and was attached using an adhesive baseplate. For lumbar mobility, the lower (sacral) sensor was positioned using a line drawn between the posterior superior iliac spines, and the upper lumbar sensor was positioned above this line using height-specific templates to ensure the accurate positioning of the upper sensor over the T12 vertebra. ViMove sensors provided an absolute orientation estimation (roll, pitch, and yaw) in real-time and calculated the relative orientation between them in three planes (frontal flexion, lateral flexion, and rotation) combining the measurements of both sensors. After placing the sensors in the cervical location, the physiotherapist guided the participants through a sequence of spinal movements—Flexion/extension, lateral flexion, and rotation—Using standardized instructions for each. All participants started in a standing position, with the same sequence of movements after a familiarization session. Once the movement was recorded, the sensors were placed in the lumbar area, and the movements were repeated three times at the lumbar level. Figure 2 shows a participant with the UCOTrack markers and the ViMove sensors.

The agreement between mobility results obtained with ViMove vs. UCOTrack was assessed in axSpA-1 participants. Figure 2 shows the location of the UCOTrack (spherical) markers and ViMove (rectangular) sensors. Both systems acquired, at the same time, different ranges of movements as maximum and minimum values in each part and in each plane (sagittal-flexion/extension, frontal-lateral bending, and axial-rotation). The data acquisition frequency of the systems was different (ViMove 12 Hz vs. UCOTrack 50 Hz). For this reason, a linear interpolation process was upsampled to 100 Hz for both signals to adjust the raw data to the same time frequency for comparing mobility curves. Appendix A shows the mobility curves for a participant obtained from both systems.

The UCOASMI score was also calculated using the IMU sensor-based system to analyze whether results similar to the UCOASMI composite score could be obtained using different technology (IUCOASMI). For discriminant and construct validity, only the IMU sensor-based system was applied in healthy participants and in the axSpA-2 group.

For the IMU sensor-based system, the angles of ROM measurement utilize the orientation components in three axes (roll, pitch, and yaw) for three planes of motion: frontal flexion, lateral flexion, and rotation. Initially, with the individual standing still, angles are recorded as zero and each IMU sensor calculates orientation angles with respect to this initial position. For example, for assessing frontal flexion, the pitch angle relative to standing is used.

Angular mobility ranges were derived from subtracting the maximum angular movement from the sensors above and below the respective regions. In this way, the range of movement of the cervical and lumbar regions was calculated. We also analyzed a third measure for the lumbar region, the angle of the L1 upper sensor to the ground (absolute orientation in the sagittal plane), representing contributions for both lumbar and pelvic movements (L1 region).

For the motion capture system, angles in the three axes were obtained by calculating the projected angles of the segment defined by two markers. For example, for cervical frontal flexion, the angle of the segment was defined by forehead and occiput markers, in the sagittal plane with respect to the ground, is measured. This angle should be very similar to the upper IMU sensor for cervical flexion. Other segments and angles are defined in UCOTrack to provide similar measures to those obtained by the IMU-based system.

Information about the setup, measures, and kinematic analysis procedures, are included in the Appendix A.

### 2.4. Sample Size and Statistical Analysis

Descriptive data are presented as the mean ± standard deviation and range (min–max) for quantitative variables and as frequencies and percentages for qualitative variables. Normal distribution was determined using the Shapiro–Wilk test, histograms, and boxplots.

Pearson’s correlation, intraclass correlation coefficient (ICC_2,1_—Two-way random effects, absolute agreement, and single rater/measurement) and Bland–Altman analysis, between the results from the IMU sensor-based and motion capture measures, were used for concurrent validity. ICC values between 0.6 and 0.8 represented a good level of agreement, and >0.8 represented excellent agreement [42]. For the Bland–Altman analysis, the mean bias, defined as the average of the differences between both systems, was determined, together with the limits of agreement (LoA), providing an estimate of the interval where 95% of the differences between both methods lie, and is defined as the bias ±1.96 standard deviations of differences. The absolute reliability was assessed using the standardized error measurement (SEM), calculated as SEM = s.d._pooled_ × √(1-ICC), and the minimum detectable change (SDC), calculated as SDC = 1.96 × SEM × √2 [43]. The unpaired Student’s t-test was used for discriminant validity testing between measures of the IMU sensor-based system and conventional metrology in the axSpA-1 and control groups. For construct validity, axSpA outcomes were correlated (Pearson) with the results obtained from the IMU sensor-based system. Values between 0.3 and 0.7 denoted weak to moderate correlation, and values above 0.7 were regarded as good correlation.

The sample size estimation (α = 5%, power = 80%) based on an anticipated ICC_2,1_ of 0.6 for concurrent validation was 18 axSpA-1 participants (ICC.Sample.Size R package), 19 axSpA-2 participants for a Pearson correlation above 0.6 for construct validity (pwr R package), and 17 participants in the axSpA-1 and control groups for significant differences in unpaired t-tests for discriminant validity (pwr R package). A two-tailed *p*-value < 0.05 was considered statistically significant, and the statistical analysis was performed using IBM SPSS software (version 17.0) and R statistical language R Studio (version 1.1.383).

## 3. Results

### 3.1. Subjects

The axSpA-1 and healthy groups comprised 20 participants each, and 50 participants from the CASTRO cohort were included in the axSpA-2 group. Table 1 shows the descriptive data of the three groups of participants. There were no significant differences between the axSpA-1 and control groups with regard to sex, age, height, weight, or BMI. Participants with axSpA showed a wide range of disease activity, function, structural change, and quality of life data in both axSpA groups.

### 3.2. Discriminant Validity

The discriminant validity was assessed by comparing the IMU measurements (Table 2) between axSpA-1 and matched healthy participants. The maximum range of movement was significantly different in the two groups (*p* < 0.05) for cervical flexion and rotation, lumbar flexion, and lumbar lateral flexion. Differences in the lumbar rotation were significant where the lumbar region was considered, but not when the trunk angle (absolute angle L1 sensor) was compared. For conventional metrology, all measures except Schober had significant differences (*p* < 0.05) between the two groups.

### 3.3. Concurrent Validity

Table 3 shows agreement results between measures obtained by the IMU sensor-based system and the motion capture system (reference standard). There was a very high level of agreement (ICC from 0.87 to 0.99) between both systems in all the measurements, except for lumbar rotation. The results of the L1 region obtained better agreement between systems (ICC from 0.91 to 0.99) than the results of the lumbar region (ICC from 0.63 to 0.93). The UCOASMI results calculated with the UCOTrack system and with the IMU sensor-based system are shown in the last row of Table 3. The agreement between the scores calculated by both systems was excellent (ICC = 0.95), with a low bias between them (−0.1 units) and low values of SEM (0.4 units) and MDC (1.1 units). These results are also expressed in Bland–Altman plots (Appendix A).

An adjusted determination coefficient R^2^ = 0.91 was obtained in a linear regression analysis between both indices (Appendix A), showing good association. In a similar analysis, after comparing these scores with BASMI, adjusted R^2^ values of 0.89 and 0.84 were obtained (Appendix A). The association between cervical rotation measured by a goniometer and by both automated systems analyzed by regression analysis also produced good results (R^2^ > 0.71) (Appendix A). 

### 3.4. Construct Validity

The correlation between the IMU sensor-based system and conventional metrology is shown in Table 4. Significant levels of correlation between measures that were conceptually related were observed, for example, lateral flexion measured with a metric tape and lateral flexion according to IMU sensors. There were good correlations between IMU lumbar region flexion and Schober (*r* = 0.66, *p* < 0.001) and between IMU and tape measure lumbar lateral flexion (*r* = 0.52, *p* < 0.001). Similar correlation results (*r* = 0.59, *p* < 0.001 and *r* = 0.51, *p* < 0.001) were determined for trunk inclination (absolute angle L1 sensor). Very good correlation was obtained for cervical rotation (*r* = 0.94, *p* < 0.001) measured by a goniometer and IMUs. The tragus-to-wall distance obtained good correlation results with cervical region mobility measures (*r* < −0.65, *p* < 0.001). There was a very strong correlation between the IUCOASMI and the conventional BASMI composite score (*r* = 0.91, *p* < 0.001).

The correlations between IMU mobility measures and other axSpA outcome variables are shown in Table 5. The BASFI had a significant correlation (*p* < 0.05) with all IMU measures (r varies from –0.33 to –0.66). There was a significant correlation between structural change (according to mSASSS) and mobility. The cervical component of the mSASSS had a higher correlation with cervical region measurements and the lumbar component with lumbar region measurements. The IUCOASMI correlated moderately well with the BASDAI (*r* = 0.40, *p* < 0.01) and strongly with the ASAS-HI (*r* = 0.55, *p* < 0.001), BASFI (r = 0.62, *p* < 0.001) and mSASSS (*r* = 0.76, *p* < 0.001).

## 4. Discussion

In the present study, we demonstrated both the accuracy (concurrent criterion validity) and practical usefulness (construct and discriminant validity) of an IMU-based method for testing spinal mobility in axSpA. A previous study using this method showed good intra- and inter-rater reliability (ICC 0.74 to 0.98 for individual spinal mobility tests, 0.96 to 0.99 for the composite scores) and good correlations with the BASMI and BASFI in a group of 40 axSpA patients [34].

Mobility results demonstrated significant differences between axSpA and healthy patients (*p* < 0.05) for almost all IMU measures (discriminant validity). There were significant correlations between IMU mobility measures and conventional metrology, and with axSpA outcome measures, especially with function and radiographic changes (construct validity).

Regarding concurrent criterion validity, results of IMU and motion capture systems show good agreement for cervical measurements (ICC > 0.87, *r* > 0.87, RMSE < 10.0°), excellent for L1 measurements (ICC > 0.91, *r* > 0.92, RMSE < 5.6°) and good agreement for lumbar measurements (ICC > 0.63, *r* > 0.69, RMSE < 14.1°). SEM was lower than 11.8° for all measures and MDC lower than 25°. Lumbar movement, especially rotation, had worse agreement due to sacral sensor rotation according to the IMU sensor (yaw) and motion capture (angle of the segment defined by left and right anterior superior iliac spine). In this case, skin motion artifacts could affect the measurement obtained by both systems. Bias was low (<10°) except for lumbar rotation.

Our results are in line with those from previous studies, in which both systems (IMU and motion capture systems) were compared with classic procedures of motion analysis, such as gait analysis and functional tests (e.g., sit-stand), obtaining good correlation coefficients (0.85 to 0.94) with small RMSE differences (2.7°–8.9°) and high ICC values (ICC > 0.96) [26].

Some studies compare agreement between both systems mounting the sensor in a plate with a cluster of markers (four or five markers) [27,28,29,32]. In this case, the sensor moves with the marker, so differences measured by RMSE, SEM, and MDC were low. Leirbekk et al., in a comparison study that included only the sagittal and coronal planes of the lumbar spine, using the same IMU sensor-based system (ViMove) in a group of healthy subjects, using a marker cluster on the sensors, obtained results very similar to those of a motion capture system (Vicon) with an RMSE lower than 2° [32].

Besides this, in our case, agreement was analyzed as total peak-to-peak ROM, not for each angular value during the movement, so this could produce higher values in terms of limits of agreements (the greatest differences will appear in these maximum values of ROM).

The UCOASMI calculated using the IMU system (IUCOASMI) had a very good correlation with both the BASMI and the classic UCOASMI, which means that both technological systems (motion capture and IMU sensors) are interchangeable in terms of a high level of agreement. However, the IUCOASMI had an even better correlation with the rest of the variables analyzed than with the individual mobility measures.

Despite the fact that both the UCOASMI and IUCOASMI systems have the potential to be interchangeable, there are several reasons to favor the IMU sensor-based system over the gait lab system for daily clinical practice and clinical research. First, dedicated facilities and expertise in gait analysis are not needed, which allows the system to be used in medical offices. Second, the results can be obtained in real-time, so doctors and patients can discuss the results at the same visit. Finally, the IMU-based procedure is shorter, partly because it is only necessary to uncover the upper part of the body. Other studies have also used IMUs instead of classic motion capture to analyze spinal mobility [28,29,30,32], however, our study is the first applied to axSpA.

Our study has several limitations and several strengths. For technical reasons related to the UCOASMI protocol, the IMU sensors and UCOTrack markers could not be placed in exactly the same location, especially on the lower back. Other studies have reported a high level of agreement locating the markers on top of the sensor. If this had been done, the accuracy of the IMU system (in terms of degrees) could have been more directly compared, but the purpose of the study was to compare the IMU system against the standard validated optical motion analysis protocol. Even though the sensors were not in precisely identical locations, we found very few differences between the results from the IMU system and those obtained with the UCOTrack system or with conventional metrology. Furthermore, the correlation of the IUCOASMI with both the classic UCOASMI and BASMI was excellent. It should also be noted that the tests were performed with patients in a standing position, while the standard ViMove protocol advises testing to be carried out in a seated position. The standing position is used as a standard for the UCOTrack and most other optical motion capture methods, and from a practical point of view, either protocol can be chosen.

In the near future, an IMU sensor-based system could be used in daily practice and clinical research as an objective tool to measure mobility in axSpA patients. In addition, this system could be used by the patients themselves to evaluate their own mobility and motivate them to engage in their treatment and monitoring. Another advantage of this system is that information can be obtained by rheumatologists through remote sharing, making it possible to remotely monitor patients in relation to pharmacological and non-pharmacological treatment approaches.

## 5. Conclusions

This study demonstrated the validity of an IMU sensor-based system for evaluating cervical and lumbar spinal mobility in individuals with axSpA. A high level of agreement with motion capture, a good correlation with axSpA outcomes, and significant differences in mobility with healthy participants were found. Future studies should address the responsiveness of this IMU-based system to complete the validation process. This type of technology represents an advance in the evaluation of patients with axSpA and can be used in future studies, both in clinical practice and in research.

## Figures and Tables

**Figure 1 diagnostics-10-00426-f001:**
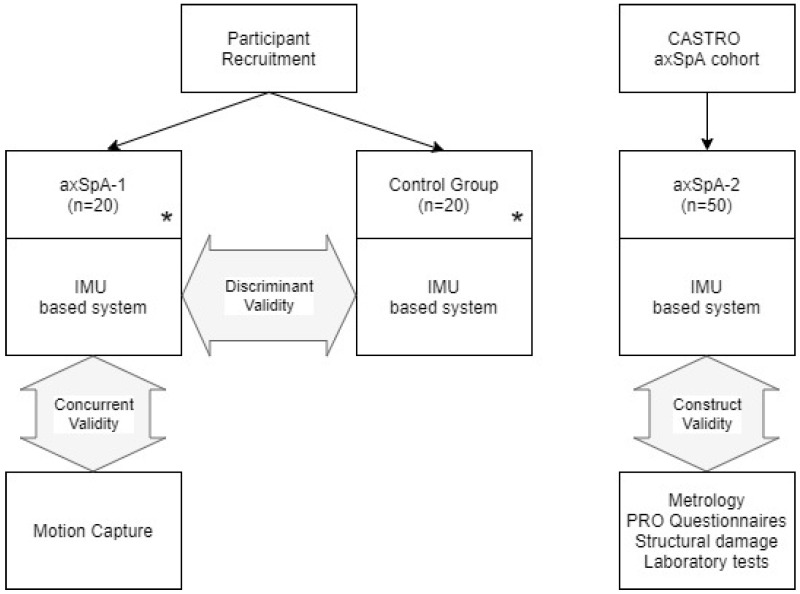
Study flow chart: Participants and measures for each validation. * age- and sex-matched groups.

**Figure 2 diagnostics-10-00426-f002:**
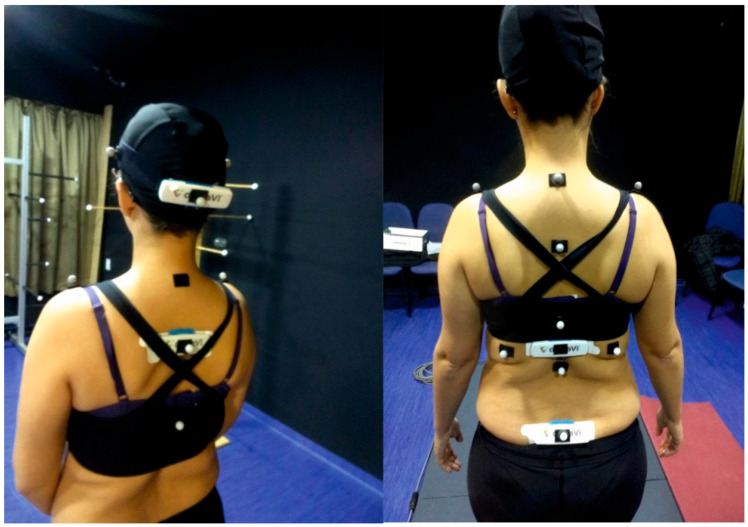
UCOTrack marker set and ViMove sensor location for analyzing cervical and lumbar spinal mobility.

**Table 1 diagnostics-10-00426-t001:** Demographics and clinical characteristics of the subjects.

	axSpA-1 (*n* = 20)		Control (*n* = 20)			axSpA-2 (*n* = 50)	
	Mean (s.d.)	Range	Mean (s.d.)	Range	*p*-Value	Mean (s.d.)	Range
Age, years	46 (12.0)	25–69	44 (8.5)	29–61	0.537	43 (12.3)	25–64
Sex (women/men)	5/15		5/15			18/32	
Disease dur., years	24 (13.7)	3–48	-	-	-	17 (13.7)	1–50
Height (m)	1.7 (0.1)	1.6–1.8	1.7 (0.1)	1.5–2.0	0.963	1.7 (0.1)	1.5–1.8
Weight (Kg)	75.1 (11.4)	51.7–100.0	75.7 (19.8)	45.3–103.6	0.915	70.6 (14.6)	48.0–125.5
BMI, kg/m^2^	25.1 (3.7)	18.9–33.8	25.0 (4.3)	17.7–32.1	0.847	26.1 (4.0)	20.1–42.4
BAS-G (0–10)	4.0 (2.9)	0.5–10	-	-	-	4.2 (2.6)	0–8
BASDAI (0–10)	3.9 (2.7)	0.4–9.4	-	-	-	3.7 (2.1)	0–8.2
BASFI (0–10)	3.2 (2.7)	0–8.8	-	-	-	2.7 (2.4)	0–9.5
ASDAS						2.4 (0.9)	0.7–4.57
mSASSS						11.8 (13.7)	0–61
ASAS-HI						4.1 (3.7)	0–12

Disease dur.: Disease duration; BAS-G: Bath Ankylosing Spondylitis Global Index; BASDAI: Bath Ankylosing Spondylitis Disease Index; BASFI: Bath Ankylosing Spondylitis Functional Index; ASDAS: ASAS Disease Activity Index; ASAS-HI: ASAS Health Index; mSASSS: Modified Stoke Ankylosing Spondylitis Spine Score. *p*-value: Student t-test; *p*-value of differences between axSpA-1 and control.

**Table 2 diagnostics-10-00426-t002:** Range of spinal movements in the study participants.

	axSpA-1 (*n* = 20)		Control (*n* = 20)			axSpA-2 (*n* = 50)	
	Mean (s.d.)	Range	Mean (s.d.)	Range	*p*-Value	Mean (s.d.)	Range
**Cervical region IMU**							
Flexion + extension (deg)	82.1 (24.8)	22–116	99.3 (15.6)	72–132	0.013 *	95.3 (23.1)	4–155
Lateral flexion L + R (deg)	56.2 (26.2)	3–104	67.7 (12.7)	45–97	0.088	64.5 (22.7)	0–111
Rotation L + R (deg)	111.9 (39.1)	36–175	138.6 (17.2)	107–163	0.010 *	129.7 (29.3)	7–180
**L1 region IMU**							
Flexion + extension (deg)	119.5 (23.6)	73–166	139.6 (19.9)	104–175	0.006 *	124.7 (23.7)	71–167
Lateral flexion L + R (deg)	50.5 (20.2)	5–77	69.5 (12.8)	45–99	0.001 *	54.8 (16.4)	10–82
Rotation L + R (deg)	126.3 (29.6)	77–204	128.2 (24.5)	70–191	0.820	138.1 (29.5)	65–185
**Lumbar region IMU**							
Flexion + extension (deg)	49.7 (21.4)	8–87	61.4 (12.5)	35–79	0.042 *	62.2 (23.4)	6–108
Lateral flexion L + R (deg)	41.8 (17.8)	3–67	53.6 (11.6)	33–74	0.018 *	44.4 (15.1)	1–68
Rotation L + R (deg)	21.2 (8.7)	6–39	26.6 (7.1)	14–39	0.038 *	25.4 (8.7)	11–43
IUCOASMI (0–10)	4.8 (1.9)	1.7–8.7	3.1 (1.0)	1.2–5.2	<0.001 *	3.9 (1.6)	0.8–9.1
**Conventional metrology**							
Side flexion L + R (cm)	11.7 (5.7)	3–21.75	21.6 (16.7)	7.25–91	0.019 *	13.9 (8.1)	2.5–52.5
Tragus-to-wall distance (cm)	13.5 (4.7)	9.5–27	11.1 (1.2)	9.25–14	0.037 *	11.4 (3.9)	1.25–28
Modified Schober (cm)	4.8 (1.9)	0.8–6.5	5.3 (1.0)	3.5–7	0.291	5.3 (1.8)	0.5–10.25
Intermalleolar distance (cm)	95.5 (16.3)	66–125	117.0 (15.4)	86–148	<0.001 *	100.0 (19.6)	50.5–138
Cervical rotation L + R (deg)	56.1 (18.1)	21–90	75.0 (7.2)	63–91	<0.001 *	65.5 (17.3)	2–91.5
BASMI_LIN_ (0–10)	3.4 (1.9)	0.8–7	1.7 (0.5)	0.9–2.6	<0.001 *	2.8 (1.7)	0.8–8.2

Cervical region: The angle between the occiput and T3 sensors. L1 region: the orientation angle from the upper L1 sensor to the ground. Lumbar region: The angle between the L1 and sacral sensors. L: Left. R: Right. IUCOASMI: Metrology index UCOASMI using measures obtained by IMU sensors. BASMILIN: Bath Ankylosing Spondylitis Metrology Index (linear). *p*-value: Student’s t-test; *p*-value of differences between axSpA-1 and control. * indicates *p* < 0.05.

**Table 3 diagnostics-10-00426-t003:** Agreement between measures obtained by the IMU sensor-based system and motion capture.

				95% LoA				
	ICC_2,1_ (95% CI)	R	RMSE	Bias	Lower	Upper	SEM	MDC
***Cervical region***								
Flexion + extension(deg)	0.982 (0.954, 0.993)	0.98	8.01	−4.2	−14.2	5.8	3.6	10.0
Lateral flexion L + R(deg)	0.874 (0.704, 0.950)	0.87	8.12	−2.1	−30.5	26.2	10.2	28.4
Rotation L + R(deg)	0.909 (0.779, 0.964)	0.91	9.99	1.2	−31.5	33.8	11.8	32.6
***L1 region***								
Flexion + extension(deg)	0.959 (0.897, 0.984)	0.96	5.59	−7.2	−20.7	6.3	4.9	13.5
Lateral flexion L + R(deg)	0.988 (0.968, 0.995)	0.99	3.74	−1.9	−8.3	4.5	2.3	6.4
Rotation L + R(deg)	0.913 (0.790, 0.966)	0.92	3.74	−0.4	−25.4	24.5	9.0	24.9
***Lumbar region***								
Flexion + extension(deg)	0.790 (0.532, 0.913)	0.81	14.11	5.5	−19.7	30.7	9.0	25.0
Lateral flexion L + R(deg)	0.939 (0.849, 0.976)	0.94	3.90	−3.0	−15.6	9.5	4.5	12.5
Rotation L + R(deg)	0.631 (0.261, 0.840)	0.69	10.36	−14.7	−32.9	3.5	6.4	17.8
IUCOASMI (0–10)	0.953 (0.882, 0.982)	0.96	0.58	−0.1	−1.2	1.1	0.4	1.1

ICC_2,1_: Intraclass correlation coefficient; r: Pearson’s correlation coefficient; RMSE: Root mean square error; 95% LoA: Bland–Altman 95% limits of agreement (deg); SEM: Standard error of the mean (deg); MDC90: Minimum detectable change (deg).

**Table 4 diagnostics-10-00426-t004:** Correlation between measures obtained by IMUs and conventional metrology.

	Lateral Flexion	TTW	mSchober	IMD	Cervical Rotation	BASMI
**Cervical region IMU**						
Flexion + extension	0.56 ***	–0.69 ***	0.51 ***	0.70 ***	0.85 ***	–0.82 ***
Lateral flexion L + R	0.44 **	–0.65 ***	0.58 ***	0.78 ***	0.71 ***	–0.77 ***
Rotation L + R	0.41 **	–0.76 ***	0.58 ***	0.64 ***	0.94 ***	–0.85 ***
**L1 region IMU**						
Flexion + extension	0.45 **	–0.44 **	0.59 ***	0.72 ***	0.58 ***	–0.72 ***
Lateral flexion L + R	0.51 ***	–0.72 ***	0.56 ***	0.64 ***	0.71 ***	–0.81 ***
Rotation L + R	0.38 *	–0.44 **	0.38 *	0.42 **	0.59 ***	–0.54 ***
**Lumbar region IMU**						
Flexion + extension	0.66 ***	–0.59 ***	0.66***	0.59 ***	0.48 **	–0.75 ***
Lateral flexion L + R	0.52 ***	–0.78 ***	0.62 ***	0.58 ***	0.67 ***	–0.81 ***
Rotation L + R	0.54 ***	–0.37 *	0.54 ***	0.43 **	0.40 **	–0.53 ***
IUCOASMI	–0.60 ***	0.71 ***	–0.68 ***	–0.77 ***	–0.83 ***	0.91 ***

Correlation coefficients between measures that are directly related are highlighted. * *p* < 0.05; ** *p* < 0.01; *** *p* < 0.001. TTW: tragus-to-wall distance; mSchober: modified Schober; IMD: intermalleolar distance.

**Table 5 diagnostics-10-00426-t005:** Correlation between measures obtained by IMUs and other important outcomes.

							mSASSS		
	Age	Dis.Dur.	BASDAI	ASDAS	BASFI	ASAS-HI	Total	Lumbar	Cervical
**Cervical region IMU**									
Flexion + extension	0.33 *	0.29	–0.41 **	–0.31 *	–0.66 ***	–0.58 ***	–0.71 ***	–0.62 ***	–0.70 ***
Lateral flexion L + R	–0.69 ***	–0.63 ***	–0.2	–0.16	–0.43 **	–0.46 **	–0.63 ***	–0.52 ***	–0.66 ***
Rotation L + R	–0.70 ***	–0.62 ***	–0.28	–0.27	–0.60 ***	–0.49 ***	–0.71 ***	–0.60 ***	–0.74 ***
**L1 region IMU**									
Flexion + extension	–0.61 ***	–0.61 ***	–0.36 *	–0.26	–0.55 ***	–0.52 ***	–0.66 ***	–0.60 ***	–0.62 ***
Lateral flexion L + R	–0.56 ***	–0.47 **	–0.25	–0.3	–0.46 **	–0.43 **	–0.67 ***	–0.66 ***	–0.57 ***
Rotation L + R	–0.45 **	–0.42 **	–0.39 *	–0.37 *	–0.45 **	–0.44 **	–0.47 **	–0.49 **	–0.37 *
**LU region IMU**									
Flexion+extension	–0.21	–0.15	–0.37 *	–0.43 **	–0.46 **	–0.47 **	–0.70 ***	–0.64 ***	–0.66 ***
Lateral flexion L+R	–0.58 ***	–0.44 **	–0.17	–0.25	–0.39 *	–0.36 *	–0.71 ***	–0.70 ***	–0.61 ***
Rotation L + R	–0.49 ***	–0.44 **	–0.33 *	–0.33 *	–0.33 *	–0.26	–0.45 **	–0.45 **	–0.38 *
IUCOASMI	0.62 ***	0.54 ***	0.40 **	0.36 *	0.62 ***	0.55 ***	0.76 ***	0.71 ***	0.71 ***

Correlation coefficients between measures directly related are highlighted. * *p* < 0.05; ** *p* < 0.01; *** *p* < 0.001. Dis.Dur.: Disease duration; BASDAI: Bath Ankylosing Spondylitis Disease Index; ASDAS: ASAS Disease Activity Index; BASFI: Bath Ankylosing Spondylitis Functional Index; ASAD-HI: ASAS Health Index; mSASSS: modified Stoke Ankylosing Spondylitis Spine Score.

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
