# Peer review of "Measuring Spinal Mobility Using an Inertial Measurement Unit System: A Validation Study in Axial Spondyloarthritis"

_diagnostics, 2020, doi:10.3390/diagnostics10060426_

Round 1

Reviewer 1 Report

This manuscript discovered concurrent criterion validity in individuals with axial spondyloarthritis (axSpA) by comparing spinal mobility measured by an IMU sensor-based system versus optical motion capture as reference standard. The study design compared discriminant validity of mobility with healthy volunteers, as well as construct validity of mobility results with relevant outcome measures. I have the following concerns for the study.

In 2.1 Study population

The study design included two groups of participant recruitment for concurrent and discrimination validations in comparison with a group of cohort measurement for construct validation. The design is ambiguous to explain the relationships between the present measurement and the cohort data. Were they following the same metrology or same status?

In 2.2 Data collection and assessment

What is the observer CGN that is not mentioned in the previous paragraphs?

In 2.3 Instrumentation

1. According to the sensor setup for the cervical and lumbar portions, there are two sensor nodes to measure the data of angles. The distance between IMU sensors could lead different relative measured values regarding angles depending on people. How to calibrating these values either by the IMU sensors or by the motion capture system? Were both systems compared by the same ranges of subjects (e.g., BMI, ROM, etc.) for calibrating their difference in measurement?

2. According to the description in Figure 1, I supposed both of ViMove and UCOTrack were monitored by the same IMU sensor-based system. It should be used for measuring the axSpA-1 group, then the results could be compared with either the control group or the cohort data. But, only the IMU sensor-based system was applied in healthy participants and in axSpA-2 group that was for a cohort study? The design statement is confused, or I mistook something?

In 2.4 Sample size and statistical analysis

If the range of motion is limited, the difference regarding the groups of patients and healthy people were probably not significant. Then, how to confirm the reliability of statistical values referring to the results in Table 1 and 2? Did the presented p-values represent the statistics for three groups, or only for the patient group and health people?

In 3. Results

Regarding the IMU sensor-based system and conventional methods, the angles of ROM measurement should involve spatial components in three axes (i.e. the orientation in roll, pitch, yaw) for three plans of frontal flexion, lateral flexion, and rotation. Which components of the angles were computed by both systems for validation? Or, the absolute values (i.e. the square root of three components) were computed? Is it possible to find the contribution of each component to the measured values?

Supplement Figure: Figure S3 in the supplemental materials cannot be opened.

Reviewer 2 Report

The paper titled “Measuring spinal mobility using an inertial measurement unit system: a validation study in axial spondyloarthritis” is aimed to assess the validity of an IMU sensor-based system vs optical motion capture system to examine the spinal mobility in axSpA. The authors selected 3 groups of subjects: a first axSpA group and a group of healthy subjects to assess discriminant and concurrent validity; a second axSpA group for analyze construct validity.

The aim of the study is clearly stated, together with the population selection criteria.

However I suggest to present the aim of the study in the introduction by focusing on the utility of IMU sensors based technology with respect to the moCap one enlarging the references with the literature. Some papers listed below could be useful for the author to better focus of the paper.

The authors state that the participants performed a sequence of movements following physiotherapist instructions. To improve the readability of the paper I suggest to describe with more details the performed motor task.

The authors should describe how they managed the IMU sensors placement and if a calibration procedure has been performed to ensure an alignment between reference frames of optical system and IMU system. Where the markers are placed? What kind of kinematic protocol has been followed? To compare kinematic data from two different motion capture systems, sensors placement and reference frames should be described.

The Discussion should be enriched comparing the results with findings present in literature. The low accuracy of IMU with respect to optical motion system recognized by authors for Lumbar region (table 3) should be discussed.

What are the limits of the study?

I suggest to take in consideration some relevant findings discussed in the following papers to improve introduction and specially the discussion:

  1. Leroy, Eve. Analysis of the spine through a multibody model and IMU technology. Ecole polytechnique de Louvain, Université catholique de Louvain, 2019. Prom. : Fisette, Paul ; Abedrabbo Ode, Gabriel. http://hdl.handle.net/2078.1/thesis:19491 Le dépôt
  2. Cardarelli S. et al. (2019) Position Estimation of an IMU Placed on Pelvis Through Meta-heuristically Optimised WFLC. In: Lhotska L., Sukupova L., Lacković I., Ibbott G. (eds) World Congress on Medical Physics and Biomedical Engineering 2018. IFMBE Proceedings, vol 68/2. Springer, Singapore
  3. Cardarelli et al., "Single IMU Displacement and Orientation Estimation of Human Center of Mass: A Magnetometer-Free Approach," in IEEE Transactions on Instrumentation and Measurement, doi: 10.1109/TIM.2019.2962295.
  4. Charles Odonkor, Anne Kuwabara, Christy Tomkins-Lane, Wei Zhang, Amir Muaremi, Heike Leutheuser, Ruopeng Sun, Matthew Smuck, Gait features for discriminating between mobility-limiting musculoskeletal disorders: Lumbar spinal stenosis and knee osteoarthritis,Gait & Posture,Vol 80, 96-100,2020,ISSN 0966-6362,https://doi.org/10.1016/j.gaitpost.2020.05.019.
  5. Juhyuk Yim, Hyunho Kim, Young-Jae Park, Young-Bae Park, A review on measuring cervical range of motion using an inertial measurement unit, The Journal of Korean Medicine 2017; 38(1): 56-71. DOI: https://doi.org/10.13048/jkm.17006
  6. Goodvin, C., Park, E.J., Huang, K. et al.Development of a real-time three-dimensional spinal motion measurement system for clinical practice. Med Bio Eng Comput 44, 1061–1075 (2006). https://doi.org/10.1007/s11517-006-0132-3
  7. Samadani, A. Lee and D. Kulić, "A Spinal Motion Measurement Protocol Utilizing Inertial Sensors Without Magnetometers," 2018 40th Annual International Conference of the IEEE Engineering in Medicine and Biology Society (EMBC), Honolulu, HI, 2018, pp. 1-4, doi: 10.1109/EMBC.2018.8512565.
  8. Mark C. Schall, Nathan B. Fethke, Howard Chen, Fred Gerr, A comparison of instrumentation methods to estimate thoracolumbar motion in field-based occupational studies, Applied Ergonomics, Vol 48, 224-31,2015, ISSN 0003-6870.

Please check some typographic inaccuracies

Table 3. Angular excursion should be expressed as deg rather than (°) and please check the extension word

Deg in Discussion (lines 295-300)

Please check the sentence in lines 234-235: “Lumbar rotation was significantly different (p=0.001) lumbar flexion but not trunk angle (absolute angle L1 sensor). Why compare lumbar rotation to lumbar flexion?...

Lines 240: “These results reflect the values obtained by IMU sensors and their equivalent in the motion capture system in the different regions analysed”. What the authors mean?

What about SEM and MDC? Can the authors discuss the obtained values for the angular parameters?

Lines 259-260: the authors should clarify what they mean when they state : “ Significant levels of correlation between measures that are conceptually related were observed”. How they establish the conceptual relation?

In “Sample size and statistical analysis” section, the authors state to use Bland-Altman plots to analyze concurrent validity. When two systems/devices are compared, the Bland-Altaman plots can provide useful indications about the agreement of measures. However I didn’t see B-A plots in the paper and in the supplementary materials. Please consider to insert the plots.

Round 2

Reviewer 1 Report

The required data were updated. My questions have been answered. I don't have more comments. The article is suggested to publish.

Reviewer 2 Report

Authors have taken into account reviewer comments and the paper is now improved in clarity and completeness.

The supplementary material has been enricched and constitute a relevant source of information to better undestand the experimental set up and some methodological aspects. Discussion contains now comparisons with findings of related literature and highlights the elements of novelty.

I suggest the authors to verify only some typos errors and consider to use deg for the Whole paper instead °.